# Exogenous dsRNA-Induced Silencing of the *Phytophthora infestans* Elicitin Genes *inf1* and *inf4* Suppresses Its Pathogenicity on Potato Plants

**DOI:** 10.3390/jof9111100

**Published:** 2023-11-11

**Authors:** Artemii A. Ivanov, Tatiana S. Golubeva

**Affiliations:** 1Institute of Cytology and Genetics SB RAS, 630090 Novosibirsk, Russia; a.ivanov2@g.nsu.ru; 2Department of Natural Science, Novosibirsk State University, 630090 Novosibirsk, Russia

**Keywords:** *Phytophthora infestans*, spray-induced gene silencing, potato late blight, exogenous dsRNA

## Abstract

*Phytophthora infestans*, an Oomycete pathogen, has a devastating impact on potato agriculture, leading to the extensive use of chemical fungicides to prevent its outbreaks. Spraying double-stranded RNAs to suppress specific genes of the pathogen via the RNA interference (RNAi) pathway may provide an environmentally friendly alternative to chemicals. However, this novel approach will require various target genes and application strategies to be tested. Using the L4440 backbone, we have designed two plasmids to express dsRNA targeting *inf1* and *inf4* genes of *P. infestans* that are known to contribute to the disease development at different stages. The dsRNA produced by the bacteria was tested on potato explants and demonstrated a statistically significant reduction in lesions five days after inoculation compared to water treatment. The study results allow us to consider our approach to be promising for potato late blight control.

## 1. Introduction

*Phytophthora infestans* is known to be one of the most challenging plant pathogens for agriculture. Every year, billions of dollars are spent to combat it, and losses in crop yields due to untimely and insufficient measures may reach up to 75% [1].

The attempts to develop a potato cultivar resistant to this pathogen based on the so-called R-genes started in the last century. However, they failed due to the high variability of the pathogen [2]. In developed countries with large farms (USA, northern Europe), the primary means of crop protection have been fungicides, with their systemic combined application providing a high level of protection and low risk of pathogen resistance [3].

However, despite the effectiveness, the massive application of fungicides raises concerns. Although the potential effects of large amounts of fungicides in the soil and groundwater are poorly explored [4], they are used in large quantities. For example, in Sweden, occupying only approximately 1% of all agricultural land, potato production consumes 21% of all fungicides in the national agricultural sector [5]. Hence, developing alternative methods to address the problem or minimize the use of chemical fungicides is imperative.

Several new approaches to reducing the damage from *P. infestans* are being actively developed in addition to applied technical solutions, such as sanitary control of tubers before planting and monitoring resistant lines by genotyping [6]. These approaches include searching for potato loci associated with quantitative resistance to *P. infestans* (QTL) [7], combining R-genes in an attempt to develop a fully resistant cultivar [8], and developing RNA interference-based biofungicides, with the last being the newest and most low-cost of the three [9,10].

Among the advantages of using exogenous RNA over conventional fungicides are high specificity, the absence of negative consequences for the environment and humans, and the possibility of quickly changing the target in response to pathogen adaptation [9]. The spectrum of target genes for silencing is broader than that of conventional fungicides. Moreover, the price of producing and applying dsRNA has equaled and even fallen below that of chemical fungicides in recent years, reaching 50 cents per gram of dsRNA in the USA [11]. The exogenous application of RNAi is a promising strategy. Its effectiveness in the future depends not only on a fundamental understanding of the mechanisms of action but also on mastering its use in practice for various plant-parasite combinations [10].

Currently, spray-induced gene silencing (SIGS) is being extensively analyzed as a way to counteract *Phytophthora infestans* [12,13,14], with dsRNA complementary to various genes being tested, including their combinations. Another popular direction is studying the role of dsRNA carriers [13], such as nanoclays [14]. Such an approach allows one to prolong the lifetime of dsRNA in the external environment and, according to some data, facilitate its entry into the plant cell [15].

In this study, two *Phytophthora infestans* genes, *inf1* and *inf4*, were subjected to silencing. The INF4 protein is actively secreted by haustoria and seems to be involved in sterol transport [16]. INF1 is involved in plant immune system modulation, causing tissue necrosis at a late stage of infection [17]. The area affected by late blight in control plants treated with water was compared with the same area in experimental plants treated with dsRNA to characterize the reduction in disease severity after dsRNA treatment.

## 2. Materials and Methods

### 2.1. Objects

Mycelium of *Phytophthora infestans* strain VZR18 was provided by the All-Russian Research Institute of Plant Protection and grown on Petri dishes with Rye B medium (“Rye B Agar”) https://www.protocols.io/view/Rye-B-Agar-36wgqj2ovk57/v1 (accessed 2 October 2022) for 30 days in the dark at +14 °C. The potato cuttings of the Nikulinsky cultivar were grown in test tubes on the MS medium (Sigma-Aldrich, Darmstadt, Germany) with the addition of 0.7% agar for 30 days at 16 h light day and 21 °C temperature.

### 2.2. Production of cDNA of inf1 and inf4 Genes

100 mg of *Phytophthora infestans* mycelium were ground in a mortar with liquid nitrogen, with QIAzol (Qiagen, Hilden, Germany) used for RNA extraction according to the manufacturer’s protocol. Reverse transcription was performed using the M-MuLV-RH reverse transcription kit (Biolabmix, Novosibirsk, Russia) according to the manufacturer’s protocol by adding 0.5 μg of total RNA as a matrix and using a random hexaprimer. PCR was performed using BioMaster HS-Taq PCR kit (2×)” (Biolabmix, Novosibirsk, Russia), with the primers used shown in Table 1. The annealing temperature was 67 °C for both pairs of primers, with the annealing time of 20 s and number of cycles of 40. Other parameters were in accordance with the manufacturer’s recommendations.

### 2.3. Production of Expression Vectors

Vectors for dsRNA expression in *E. coli* were assembled based on plasmid L4440 L4440 (Addgene plasmid #1654; RRID: Addgene_1654). Plasmid L4440 and cDNA were hydrolyzed using PstI and Bsp19I restriction endonucleases (Sibenzyme, Novosibirsk, Russia). The reaction was performed according to the manufacturer’s protocol, with an incubation time of 16 h. The reaction products were purified by Evrogen Cleanup S-Cap kit (Evrogen, Moscow, Russia).

Ligation was performed using T4 DNA ligase (Evrogen, Moscow, Russia) with a proprietary buffer. The reactions were conducted for 16 h at 4 °C, with a plasmid-to-insert ratio of 1:5. Transformation was performed by the standard CaCl2 method, with seeding on medium containing ampicillin (100 μg/mL) and tetracycline (12.5 μg/mL). The insert in the colonies was verified by PCR under the conditions described above, followed by the isolation of plasmids from positive colonies using a Plasmid 250-mini kit (Biolabmix, Novosibirsk, Russia). The plasmids were sequenced by Sanger sequencing at the Genomics Center of ICBFM SB RAS. The resulting maps are presented in Figure 1.

### 2.4. Production of dsRNAs

For obtaining dsRNAs from the colonies containing L4440 carrying the *inf1* or *inf4* insert, overnight cultures were prepared in 5 mL of LB medium with ampicillin (100 μg/mL) and tetracycline (12.5 μg/mL). In the morning, the fresh overnight culture was transferred to 50 mL of LB medium with ampicillin (100 µg/mL) and incubated at 37 °C under continuous stirring at 200 rpm until OD_600_ = 0.5. Next, IPTG (Thermo Fisher Scientific, Waltham, MA, USA) was added to the culture to a concentration of 0.6 mM to induce dsRNA production and then left to grow under the same conditions for 4 h.

For preparing purified dsRNA, 50 mL of bacterial culture was precipitated by centrifugation (5 min, 5000× *g*) and resuspended in 5 mL of a 1:1 mixture of 1 M ammonium acetate and 10 mM isoamyl acetate. A mixture (5 mL) of phenol, chloroform, and isoamyl acetate (25:24:1) was added to the suspension and incubated for 30 min at 65 °C with occasional stirring. The mixture was then centrifuged for 20 min at 4 °C and 12,000× *g*, the aqueous phase was withdrawn, 1 volume of isopropanol was added, and the mixture was left overnight at −20 °C. The next day, centrifugation was performed for 10 min at 4 °C 12,000× *g*, and the resulting precipitate was washed twice with 70% ethanol, followed by centrifugation for 2.5 min at 4 °C 12,000× *g*. The precipitate was dried at 37 °C until the alcohol was completely removed and then dissolved in 100 μL of RNase-purified water.

Next, the treatment with DNase (Thermo Fisher Scientific, Waltham, MA, USA) and then with RNase A (Sigma-Aldrich, Darmstadt, Germany) was performed. DNase treatment was performed according to the manufacturer’s protocol, with incubation at 37 °C for 30 min. RNase A treatment was performed in 0.3 M NaCl solution to exclude dsRNA hydrolysis, with 0.5 µL of 10 mg/mL RNase A solution added per 100 µL of the reaction mixture, followed by incubation for 15 min at 37 °C.

After treatment, dsRNA was extracted from the reaction mixture using a phenol/chloroform/isoamyl acetate mixture, followed by isopropanol re-precipitation and double ethanol washing as described above.

### 2.5. Treatment and Inoculation of S. tuberosum Plants

One-month-old explants of *S. tuberosum* cultivar Nikulinsky were treated with purified dsRNA at a concentration of 100 ng/μL using a micropipette at the dose of 5 μg per plant. The dsRNA was applied to the adaxial surface of the lower leaves (5 μL per leaf, 10 leaves per plant) or mixed with the nutrient medium (50 μL at 100 ng/μL). Distilled water was used as a control. A group of two plants was used for each of the following treatments:(1)*inf1* dsRNA was applied to the leaves;(2)*inf1* dsRNA was added to the media;(3)*inf4* dsRNA was applied to the leaves;(4)*inf4* dsRNA was added to the media;(5)full doses of *inf1* and *inf4* dsRNA were applied to the leaves simultaneously;(6)full doses of *inf1* and *inf4* dsRNA were added to the media simultaneously;(7)distilled water was applied to the leaves and added to the media.

The inoculation was performed 24 h after treatment. The mycelium of *P. infestans* was incubated for 3 h in distilled water at +4 °C and then filtered through a sterile gauze. Luna Cell Counter (Logos Biosystems, Anyang, Republic of Korea) was used to count the concentration of zoospores. Zoospores were applied with a micropipette to the abaxial side of the leaf at a rate of 1500 per leaf. The lesion assessment was done after 5 days.

### 2.6. Statistical Analysis

For statistical processing, each leaf was assigned a rank by the lesion area, with 9 representing a completely healthy leaf and 1 representing a completely infested leaf, as described in [18]. The choice of the scoring system is due to the small size of the leaves and the large number of specimens with 9, 8, and 1 ranks that were easy to differentiate visually. Controversial cases were measured using a binocular (16× magnification) and processed in the Image J program. Both the Mann-Whitney test and Student’s *t*-test were used to determine the significance of the effect of dsRNA treatment on potato protection against *P. infestans*. Simultaneous application of these methods was necessary because of numerous duplicate values. The calculations were performed in R 4.2.2 using the package tidyverse 2.0.0 [19].

## 3. Results

### 3.1. Production of dsRNA-Producing Strains

For the production of dsRNAs, the fragments of the *inf1* and *inf4* genes were inserted into the plasmid L4440 between the two late promoters of phage T7 at the *Nco*I and *Pst*I sites. Figure 1 presents the maps of the resulting plasmids. The sequences of both insertions were validated by Sanger sequencing and contain no discrepancies with the reference sequence.

### 3.2. Production of dsRNAs

Phenol-chloroform extraction was used for dsRNA isolation from *E. coli* HT115 to obtain the required amount of dsRNA: 250–500 µg from 50 mL of culture. Verification of the obtained RNA as dsRNA requires undertaking the RNase A purification step. This is because dsRNA displays greater resistance to hydrolysis by RNase A at NaCl concentrations above 0.3 M, compared to single-stranded RNA [20]. Furthermore, although reducing the yield, purification allows a more accurate estimation of the dsRNA concentration after the removal of genomic DNA and single-stranded RNA residues. The presence of dsRNAs in the sample after two-step purification (DNase + RNase A) was confirmed by electrophoretic analysis (Figure 2).

### 3.3. Plant Inoculation

The inoculation with *P. infestans* was carried out one day after the dsRNA treatment. The zoospore mobility was preliminarily checked using a light microscope. Throughout the experiment, the plants were kept in test tubes, providing not only isolation but also the high humidity necessary for inoculation.

The experimental results were recorded after five days, with apparent differences visually observed between control and experimental plants (Figure 3).

### 3.4. Statistical Processing of the Results

All the plants treated with dsRNA both through the leaf surface and the root system demonstrated significant protection against late blight compared to the control (Table 2). No significant differences in protection efficacy were found for the various methods of treatment and preparations.

Since no statistically significant differences were found between leaf treatment and uptake from the basal stem, the group data were combined in the graph (Figure 4). Thus, the dsRNAs of the *inf1* and *inf4* genes were found to protect potatoes against *Phytophthora infestans* regardless of the route of entry into the plant.

## 4. Discussion

Exogenous dsRNA-induced gene silencing is a promising approach to plant protection that may supplement or even replace the conventional fungicide treatments [21] that are harmful to the environment [4]. Since plants and pathogens can have unique characteristics [22] affecting the strategies of the approach concerned [23], one should carefully consider exogenous dsRNA application for each host-pathogen pair and test different solutions.

Currently, there is no consensus in the literature regarding the effectiveness of the exogenous dsRNA application in combating *Phytophthora infestans*. The direct capture of dsRNA by the pathogen from the external environment was investigated in [23], and it was concluded that this behavior is not characteristic of *P. infestans*. In contrast, the protective effect of exogenous dsRNA against *P. infestans* was demonstrated in [12]. Both investigations involved inoculation of the dsRNA-treated site with *P. infestans*. However, in the first case, dsRNA application and infection were not temporally separated, which may have had a negative effect on the protective effect.

In our study, infection and treatment were separated not only in time but also in space (lower and upper sides of the leaf, respectively), with results indicating that it is through the plant that dsRNA enters *P. infestans*. Our approach is closer to the field conditions, with the literature describing the cases of increased protection efficiency when dsRNA passes through the host cells rather than being locally exposed at the point of inoculation [22]. At the same time, the route of dsRNA entry into the plant in our experiment turned out to be unimportant: plants were equally protected when getting dsRNA from the leaf surface or through the lower stem and roots.

Sanju et al. [24] demonstrated earlier that suppression of a single effector gene is insufficient to protect potatoes against *P. infestans*. In contrast, our results indicate that, given a moderately resistant cultivar, reducing the expression of even one effector can suffice to reduce the disease severity. However, we failed to detect any difference between the suppression of single effectors or their combination, allowing us to assume that the plant’s resistance does play a significant role in this case.

In summary, preparations that inhibit specific effectors, such as *inf1* or *inf4,* can be used to enhance the protection of potato plants against late blight. Using such specific target genes can reduce the number of potential off-target effects directed at the soil organisms whose genomes and transcriptomes are not yet in databases and cannot be taken into account during exogenous dsRNA application design.

## Figures and Tables

**Figure 1 jof-09-01100-f001:**
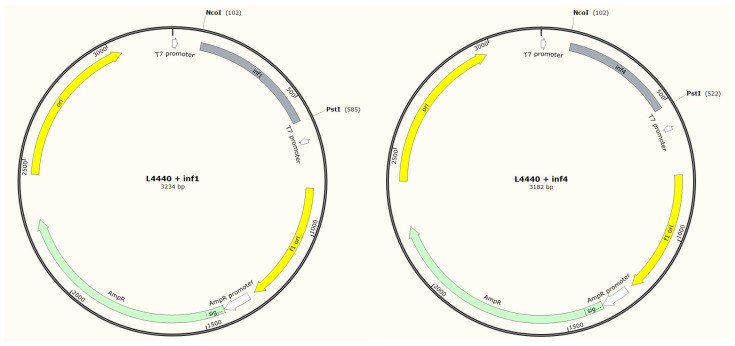
Maps of the plasmids obtained with the *inf1* and *inf1* inserts.

**Figure 2 jof-09-01100-f002:**
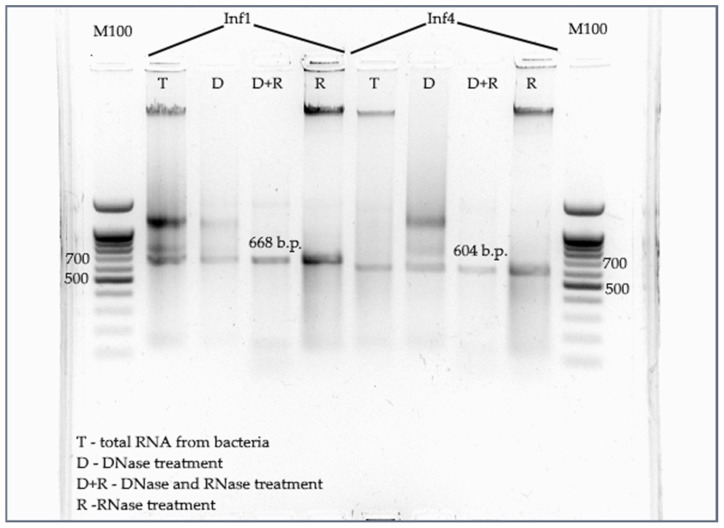
Electrophoretic analysis of dsRNA fragments synthesized in *E. coli* HT115 treated with DNase or RNase A. The agarose concentration is 1.5%, the intensity is 10 V/cm, and the time is 45 min. DNase treatment included incubation for 30 min at 37 °C, and RNase treatment included a 15 min incubation at the same temperature. Aliquots containing 500 ng of nucleic acid each were used for treatment and subsequent application to the gel.

**Figure 3 jof-09-01100-f003:**
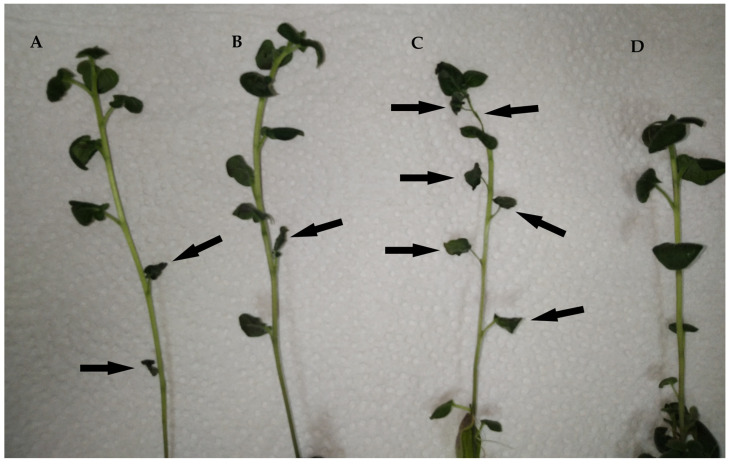
Visual differences between treated, untreated, and healthy plants (microscope magnification 1×). (**A**)—treated with *inf1* dsRNA, (**B**)—*inf4* dsRNA-treated plant, (**C**)—water-treated plant, (**D**)—not inoculated plant. Arrows indicate withered parts and leaves ranked 1 (the lowest) protection rank.

**Figure 4 jof-09-01100-f004:**
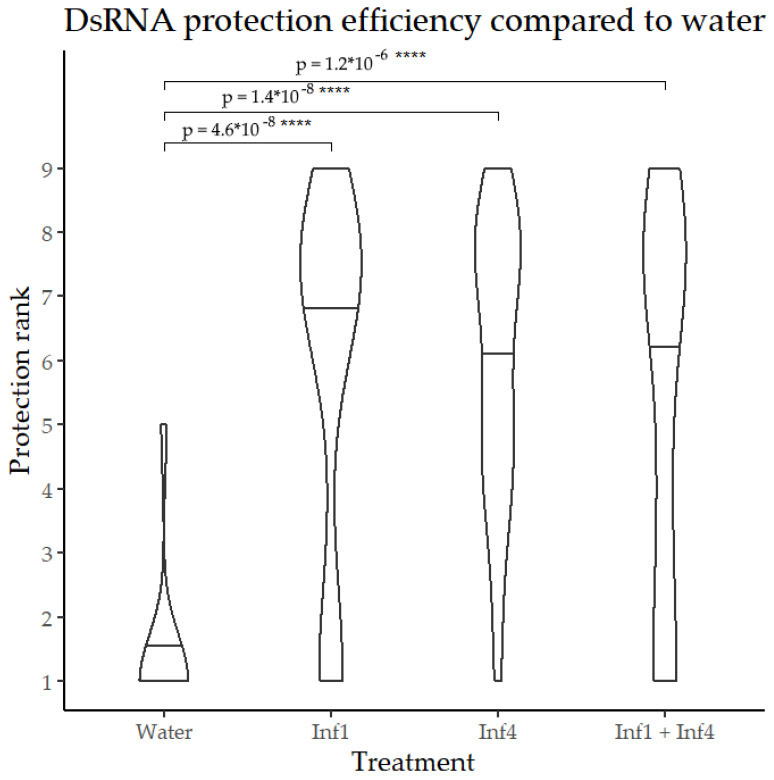
Potato protection efficiency against *Phytophthora infestans* by the dsRNA treatment. Irrigation and leaf dsRNA treatments are combined. Horizontal lines represent median values. The *y*-axis shows the protection rank calculated according to [18], with 1 indicating more than 90% lesions of a leaf caused by *P. infestans* and 9 signifying no lesions. A *t*-test was used for pairwise comparisons with water, **** marks *p*-values below 10^−4^.

**Table 1 jof-09-01100-t001:** Primers for amplifying the fragments of *inf1* and *inf4* genes.

Gene, NCBI Database Identifier, Product Length	5′-3′ Sequence	Restriction Endonuclease Site	Annealing Temperature
*inf1*—U50844.1 496 bp	ACACTACTGCAGTAATAGCAACAGACGCGGAG	Pst I	67.1 °C
	CCACTCCCATGGACTCCGTCCACGATGAACTTT	Bsp19 I	66.8 °C
*inf4*—XM_002895013.1 435 bp	CCATTGCTGCAGACAACCACTTCATCCAGCACA	Pst I	65.9 °C
	GATTCATCCATGGTATGGGATTGCAGACATGCCG	Bsp19 I	66.7 °C

**Table 2 jof-09-01100-t002:** Potato protection efficiency against *Phytophthora infestans* for different dsRNA and treatment combinations (experiment 1).

Method of Treatment	Effectiveness of Protection *	Number of Leaves	Significance Level *p* **	Significance Level *p* ***
Water	1.4 ± 0.41	10	-	-
*Inf1* leaf treatment	5.5 ± 0.66	21	0.000006	0.006
*Inf1* uptake from the basal stem	6.8 ± 0.71	10	0.000005	0.0008
*Inf4* leaf treatment	5.9 ± 0.62	15	0.000002	0.0003
*Inf4* uptake from the basal stem	6.1 ± 0.58	11	0.000002	0.0003
*Inf1+inf4* leaf treatment	5.9 ± 0.71	15	0.000008	0.0008
*Inf1+inf4* uptake from the basal stem	4.9 ± 0.87	14	0.001005	0.015

* Leaf lesions were ranked from 9 to 1, with 9 indicating a completely healthy leaf and 1 indicating a completely dead leaf. For the details, one is referred to [18]. The mean ± standard error of the mean is indicated. ** Student’s *t*-test for one-sided distribution, with *p*-value indicated compared with the result of water treatment. *** Mann-Whitney test, with *p*-value indicated compared with the result of water treatment.

## Data Availability

No new data were created or analyzed in this study. Data sharing is not applicable to this article.

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
