# Peer review of "Exogenous dsRNA-Induced Silencing of the Phytophthora infestans Elicitin Genes inf1 and inf4 Suppresses Its Pathogenicity on Potato Plants"

_jof, 2023, doi:10.3390/jof9111100_

Round 1

Reviewer 1 Report

Comments and Suggestions for Authors

In this manuscript, the authors have developed plasmids to produce dsRNAs targeting two genes from P. infestans, inf1 and inf4. They have furthermore shown that dsRNAs generated using these two plasmids, when applied to leaves or roots of potato plants prior to inoculation with P. infestans, greatly reduce the severity of late blight symptoms. This research lays the groundwork for an environmentally-friendly strategy of controlling this devastating pathogen, and it also clarifies the mechanism by which such sprays can work. Since the dsRNAs are applied at different physical locations on the plant and and at different times compared to the inoculation with P. infestans, the pathogen must be taking up the dsRNAs via the plant cells rather than directly from the environment.

I have only a few small suggestions to improve clarity in this manuscript.

I suggest referring to Fig. 1 in section 2.3 where you discuss the creation of the expression vectors.

Line 85 - Italicize E. coli

Line 129-130 - "The dsRNA was distributed to the lower leaves (5 μl per leaf, 10 leaves per plant) or applied to the nutrient medium (50 μl at 100 ng/μl). " The dsRNA was applied to the adaxial surface of the leaf, correct? It would help to add that here in the methods section.

Table 2 - Please put in the table legend that the table indicates how many plants were used per treatment. Also, for clarity, please define "irrigation" in the table legend.

Line 130 - I'm assuming the MS medium was liquid, not containing added agar, and you added the dsRNA and mixed it? I would consider saying it was mixed into the medium rather than "applied" to it

Author Response

Response letter

We appreciate the Reviewer’s comments. We have carefully addressed all the criticism and made the necessary changes to the manuscript. Below, we provide our point-by-point answers.

Comments to the Authors

  1. I suggest referring to Fig. 1 in section 2.3 where you discuss the creation of the expression vectors.

We appreciate the suggestion and have added a reference to Figure 1 in the appropriate section. Also, we have relocated Figure 1 in the text for better readability.

  1. Line 85 - Italicize  coli

We apologize for this inaccuracy. It has been corrected.

  1. Line 129-130 - "The dsRNA was distributed to the lower leaves (5 μl per leaf, 10 leaves per plant) or applied to the nutrient medium (50 μl at 100 ng/μl). " The dsRNA was applied to the adaxial surface of the leaf, correct? It would help to add that here in the methods section.

We are grateful for the comment. We have specified the side of the leaf in the methods paragraph.

  1. Table 2 - Please put in the table legend that the table indicates how many plants were used per treatment. Also, for clarity, please define "irrigation" in the table legend.

We have corrected the table heading and replaced irrigation with a more appropriate term.

  1. Line 130 - I'm assuming the MS medium was liquid, not containing added agar, and you added the dsRNA and mixed it? I would consider saying it was mixed into the medium rather than "applied" to it

We recognize that there is some awkward wording in this paragraph and would like to apologize for any confusion it may have caused. The MS medium in our experiment contained agar at a concentration of 0.7%. However, we agree that the word mixed does fit better to understand the process described, taking into account that the medium was softened by the roots during the month of growth. We changed the verb to a more appropriate term and added information about the density of the medium in section 2.1.

With kind regards, the Authors.

Reviewer 2 Report

Comments and Suggestions for Authors

This work is interesting and the experiments were performed  correctly, although for me it is simply not enough work/experiments/informations  for a journal with  IF 4.7. Moreover, the number of infected plants is small, so the sample is not very representative. Why is the number of plants treated with water only 2? In several places in the text, the name of the bacteria is not written in italics.

Author Response

Response letter

We are grateful for the Reviewer’s time and effort dedicated to providing the feedback. We appreciate all the comments that allow us to refine our manuscript.

Comments to the Author

This work is interesting and the experiments were performed correctly, although for me it is simply not enough work/experiments/informations for a journal with IF 4.7.

Authors’ response to the comments:

Dear Reviewer, we would like to express our gratitude for your appreciation of our study and the soundness of the experiment’s design. It is a great honor to receive such a comment from a reviewer. Our choice of the Journal is primarily due to the highly positive experience of interacting with the editorial board and reviewers. After receiving your review, we are now once again very sure of the correctness of our choice. The high impact factor of the Journal of Fungi is obviously also a solid argument for submitting a manuscript.  

Also, we are deeply grateful for your patience and helpful comments aimed at eliminating any discrepancies in the manuscript. We hope our honest answers to your comments on staging the experiment will clear up several concerns about our work.

Comments to the Authors

  1. Moreover, the number of infected plants is small, so the sample is not very representative. Why is the number of plants treated with water only 2?

We completely agree with your comment. A larger sample size is always preferable. Unfortunately, we could not afford to place a large number of plants of the same age in the same conditions at our facility. The sampling was made as large as possible. It should be noted that, even under our conditions, we have already seen a reliable protective effect of double-stranded RNA. Also, due to some experimental limitations, we decided to increase the number of plants in the experimental groups by sacrificing the number of plants for the water treatment. This was done because the likelihood of getting any outstanding results from treating potatoes with water is minimal.

  1. In several places in the text, the name of the bacteria is not written in italics.

We apologize for this inaccuracy. The manuscript has been carefully checked for italics for Latin terms.

In conclusion, we hope that our arguments and corrections, as well as the opinion of Reviewer 1 about the high scientific value of our work, will be taken into account by you when making a decision.

With kind regards, the Authors

Reviewer 3 Report

Comments and Suggestions for Authors

The manuscript by Artemii et al described the application of SIGS of inf1 and inf4 and investigated the effects of this method on plant protection.

Manuscripts is well written with a clear structure. Overall, this is a decent paper that merit the publication. 

I only have some minor comments for further consideration.

Minor comments:

Figure2: Are sample of Inf1 D and D+R that used for loading agarose gel isolated from the reaction mix? What about the loading amount? If they are isolated reaction mix with same amount, why target dsRNA amount seems increased after the treatment?

Figure3: Why untreated plant looks shortest?

Comments on the Quality of English Language

 Line 215-217: Please rephrase the sentence.

Line 231-232: Please rephrase the sentence.

Author Response

Response letter

We appreciate the Reviewer’s comments. We have carefully addressed all the criticism and made the necessary changes to the manuscript. Below, we provide our point-by-point answers.

Comments to the Authors

  1. Figure2: Are sample of Inf1 D and D+R that used for loading agarose gel isolated from the reaction mix? What about the loading amount? If they are isolated reaction mix with same amount, why target dsRNA amount seems increased after the treatment?

We appreciate the comment. We have added the information about the loading amount in the description of Figure 2. Unfortunately, this gel is not well suited for quantification. We provide these data as evidence for the qualitative presence of dsRNA since some factors can affect visualization. In particular, the manufacturer states that the buffer supplied with DNAase may lead to smearing, and the high concentration of NaCl observed only in the D+R and R lanes may also have an unspecified effect.

  1. Figure3: Why untreated plant looks shortest?

Plants can grow at slightly different rates on nutrient media. The age of the plants in this photo is the same, but the untreated plant is slightly shorter than the others. We regret not having chosen the most relevant of untreated plants for the photo.

With kind regards, the Authors.
